# Potassium channels act as chemosensors for solute transporters

Rían W. Manville[1] & Geoffrey W. Abbott[1✉]

Potassium channels form physical complexes with solute transporters in vivo, yet little is known about their range of possible signaling modalities and the underlying mechanisms. The KCNQ2/3 potassium channel, which generates neuronal M-current, is voltage-gated and its activity is also stimulated by binding of various small molecules. KCNQ2/3 forms reciprocally regulating complexes with sodium-coupled *myo*-inositol transporters (SMITs) in mammalian neurons. Here, we report that the neurotransmitter γ-aminobutyric acid (GABA) and other small molecules directly regulate *myo*-inositol transport in rat dorsal root ganglia, and by human SMIT1-KCNQ2/3 complexes in vitro, by inducing a distinct KCNQ2/3 pore conformation. Reciprocally, SMIT1 tunes KCNQ2/3 sensing of GABA and related metabolites. Ion permeation and mutagenesis studies suggest that SMIT1 and GABA similarly alter KCNQ2/3 pore conformation but via different KCNQ subunits and molecular mechanisms. KCNQ channels therefore act as chemosensors to enable co-assembled *myo*-inositol transporters to respond to diverse stimuli including neurotransmitters, metabolites and drugs.

[1] Bioelectricity Laboratory, Department of Physiology and Biophysics, School of Medicine, University of California, Irvine, CA, USA. ✉email: abbottg@hs.uci.edu

K$^+$ efflux through K$^+$-selective channels, including voltage-gated potassium (Kv) channels, represents a major inhibitory force in excitable cells, regulating resting membrane potential, terminating action potentials, and tuning firing frequency. In the neurons of the majority of multicellular organisms, including vertebrates, the M-current, a muscarinic-inhibited Kv current[1] generated by KCNQ (Kv7) family Kv channels (M-channels)[2,3], is required for control of neuronal excitability[4]. The M-current is generated primarily by KCNQ2 and KCNQ3 subunits, predominantly in the form of heteromeric (KCNQ2/3) channels[4], but homomers also exist in vivo and KCNQ4 and KCNQ5 also contribute[5]. KCNQ2/3 channels are muscarinic acetylcholine receptor-regulated, non-inactivating, and open at relatively hyperpolarized membrane potentials. They are therefore uniquely placed to regulate neuronal excitability. Hence, KCNQ2/3 gating state (open versus closed) dictates phasic versus tonic firing of neurons[4].

We recently discovered a novel form of regulation for some neuronal KCNQ channels. The primary inhibitory neurotransmitter in higher animals, γ-aminobutyric acid (GABA) is well-known for its essential role in inhibitory neurotransmission, whereby it activates canonical GABA receptors including GABA$_A$ receptors (GABA$_A$Rs), which are ligand-gated, chloride-selective channels that open to suppress neuronal excitability[6]. The other major canonical class of GABA receptors, GABA$_B$Rs, are G-protein-coupled receptors that, upon GABA binding, activate other proteins, including certain types of K+ channel (but not KCNQs)[7]. Unexpectedly, we found that some members of the KCNQ Kv channel subfamily (isoforms 2–5), which bear no resemblance in structure or topology to canonical GABARs, can also directly bind GABA, via a tryptophan residue on transmembrane segment 5 (S5). Furthermore, we found that in KCNQ3, KCNQ5, and KCNQ2/3 heteromers, GABA binding results in channel activation, and thus cellular hyperpolarization[8].

Because some KCNQ channels are sensitive to physiologically relevant (submicromolar) concentrations of GABA, GABA activation of KCNQ channels therefore presents an additional mechanism for GABA-dependent inhibition in the CNS. Interestingly, KCNQ2 subunits are enriched in GABAergic neurons, in particular those important for regulating network oscillations and synchrony[9]. Presynaptic KCNQ2/3 channels can modulate glutamate and GABA release, and may act pre- and post-synaptically to suppress neuronal excitability[10,11]. Further, *Kcnq2* deficient mice exhibit abnormal GABAergic synaptic transmission[12]. Vertebrate nervous systems also contain many GABA analogues and metabolites, some of which do not modulate canonical GABARs, but can activate select KCNQ isoforms. One example is β-hydroxybutyrate (BHB), a ketone body that accumulates during ketosis and is thought to underlie antiepileptic effects of fasting and ketogenic diets; another is γ-amino-β-hydroxybutyric acid (GABOB), which acts as a KCNQ2/3 high-affinity partial agonist[8].

KCNQ2/3 channels can form complexes with sodium-coupled *myo*-inositol transporters SMIT1 and SMIT2[13,14], which use the downhill Na$^+$ gradient to transport into cells *myo*-inositol, an important osmolyte and precursor of cell signaling molecules including phosphatidylinositol 4,5-bisphosphate (PIP$_2$), which is highly influential in the gating of KCNQ and other ion channels[15–18]. We previously detected complex formation between KCNQ2, KCNQ3, SMIT1, and/or SMIT2 in mouse neuron axon initial segments, and in rat and mouse sciatic nerve nodes of Ranvier[14]. SMIT1 and SMIT2 each negatively shift the voltage dependence of KCNQ2/3 channel activation, with superficially similar effects to GABA activation. This effect of SMIT1 and SMIT2 on KCNQ2/3 voltage dependence occurs by physical interaction with the KCNQ2/3 channel pore, without the need for *myo*-inositol. However, addition of *myo*-inositol further augments KCNQ2/3 activation when in complexes with SMIT1

(or SMIT2) by providing a substrate for local production of PIP$_2$. In addition, SMIT1 co-assembly alters the pharmacology of KCNQ2 and KCNQ2/3 channels[13,14].

Here, we report that GABA and its metabolites can modulate *myo*-inositol uptake of SMIT1 via chemosensing by co-assembled KCNQ2/3, and we identify molecular mechanistic determinants of this cross-talk, uncovering a, to our knowledge, novel paradigm for cellular signaling.

## Results

**GABA and related metabolites regulate SMIT1 via KCNQ2/3.** KCNQ channels, like all Kv channels, form as tetramers of α subunits, each subunit of which contains six transmembrane (S) segments, split into the voltage-sensing domain (VSD, S1–4) and the pore module (S5–6). SMIT1 and SMIT2 transporters have a predicted 14 transmembrane segment topology (Fig. 1a). As we previously found that KCNQ2/3 channels co-localize with SMIT1 in rodent neurons and co-assemble with SMIT1 in vitro[14], we first quantified the effects of GABA on *myo*-inositol uptake of whole dorsal root ganglia (DRG) isolated from postnatal day 8 rats. GABA (100 μM) reduced [$^3$H]*myo*-inositol uptake >twofold (Fig. 1b). In *Xenopus* oocyte expression studies, heterologously expressed SMIT1 activity was insensitive to GABA (100 μM) when expressed alone (Fig. 1c). In contrast, when co-expressed with KCNQ2/3, SMIT1 activity was inhibited >twofold, replicating our observations from rat DRG. Glutamate, which has no effect on KCNQ2/3 channels[8], had no effect on SMIT1 activity in the presence of KCNQ2/3. However, β-hydroxybutyrate (BHB), γ-aminohydroxybutyrate (GABOB), and the anticonvulsant retigabine, each of which, like GABA, activates KCNQ2/3 by negative shifting its voltage dependence of activation[8], each inhibited SMIT1 activity in the presence of KCNQ2/3 (Fig. 1d). We previously found that an S5 tryptophan conserved in neuronal KCNQs (e.g., W265 in KCNQ3) is required for binding to and activation of KCNQs by GABA[8]. Here, mutation to leucine of KCNQ2–W236 and KCNQ3–W265 in KCNQ2/3 channels prevented GABA inhibition of co-expressed SMIT1 activity (Fig. 1e). Thus, GABA binding to the S5 tryptophan of KCNQ2/3 channels inhibits *myo*-inositol uptake by co-expressed SMIT1.

**SMIT1 alters small molecule effects on KCNQ2/3 gating.** Heterologous co-expression in *Xenopus laevis* oocytes of human KCNQ2/3 channels and SMIT1 generated slow-activating, slow deactivating, non-inactivating voltage-dependent K+ currents, which we measured using two-electrode voltage clamp (TEVC) (Fig. 2a). Bath application of GABA (10 nM–10 mM) increased the peak tail currents, negative shifted the voltage dependence of activation of KCNQ2/3 channels (Fig. 2a, b) and increased activation rate (Fig. 2c), as we previously found for GABA effects on KCNQ2/3 in the absence of SMIT1[8]. SMIT1 did not alter the maximal effects of GABA on KCNQ2/3 currents; however, SMIT1 co-expression reduced the potency of GABA with respect to KCNQ2/3 activation threefold, as measured by current fold increase at −60 mV, where changes are higher than at more depolarized voltages but accurately quantifiable (KCNQ2/3, EC$_{50}$ = 1.1 ± 0.39 μM GABA; KCNQ2/3–SMIT1, EC$_{50}$ = 3.0 ± 0.67 μM GABA) (Fig. 1d). We observed an approximately similar (fourfold) reduction in KCNQ2/3 GABA sensitivity upon SMIT1 co-expression when we quantified the shift in voltage dependence of activation ($\Delta V_{0.5\text{activation}}$) (KCNQ2/3, EC$_{50}$ = 137 ± 23 nM GABA; KCNQ2/3–SMIT1, EC$_{50}$ = 593 ± 125 nM GABA) (Fig. 1e).

KCNQ2/3–SMIT1 complexes were also sensitive to BHB, as we previously observed for KCNQ2/3; in the presence of SMIT1, BHB increased the peak tail current, negative shifted the voltage dependence of KCNQ2/3 activation (Fig. 2g) and speeded

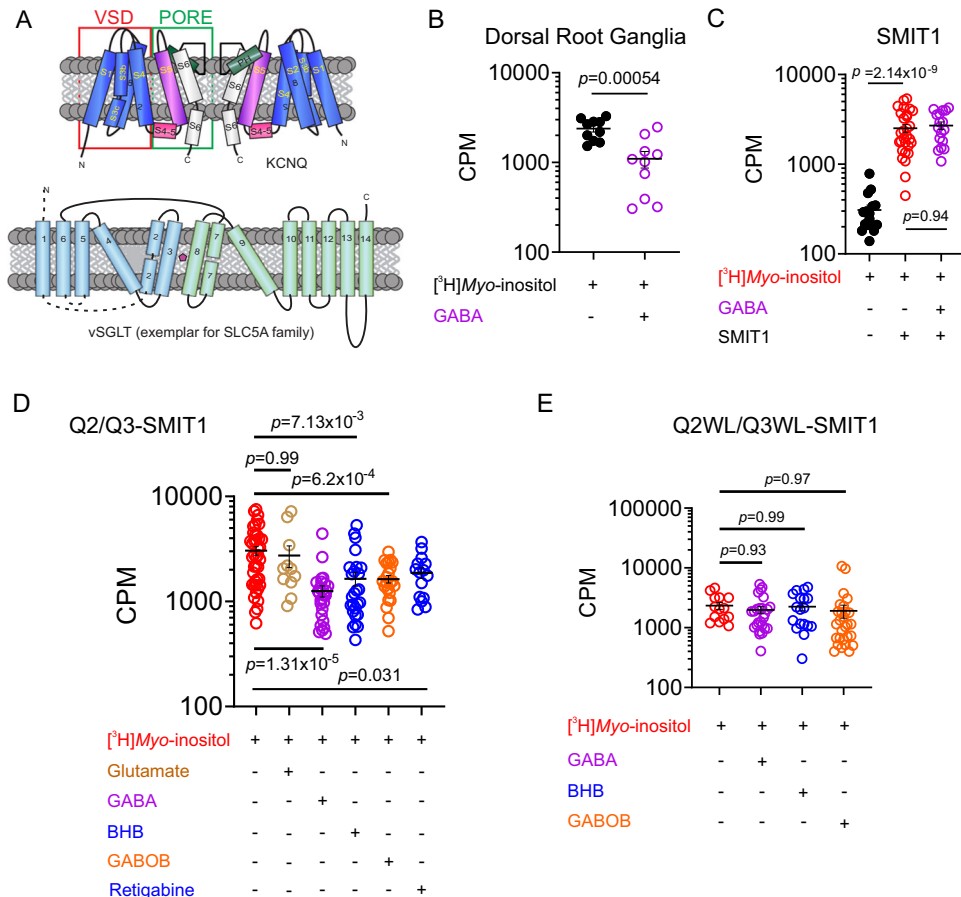

**Fig. 1 GABA regulates SMIT1 activity via co-assembled KCNQ2/3.** All error bars in the study indicate SEM. All n values in the study represent biologically independent samples or experiments. **a** Topology of KCNQ channels (two of four subunits shown) and an exemplar of the SLC5A solute transporter family, vSGLT, which may reflect SMIT topology. **b** Effect of GABA (100 μM) on [3H]Myo-inositol uptake by whole dorsal root ganglia (n = 10) isolated from postnatal day 8 rats. CPM = counts per minute. **c** Effects of GABA (100 μM; n = 17) on [3H]Myo-inositol uptake by SMIT1 (n = 29 without GABA) expressed alone in Xenopus oocytes. Control oocytes (black) were injected with water instead of SMIT1 cRNA (n = 16). **d** Effects of small molecules as indicated (100 μM) on [3H]Myo-inositol uptake by SMIT1 co-expressed with KCNQ2/KCNQ3 subunits in Xenopus oocytes ([3H]Myo-inositol n = 41; Glutamate n = 11; GABA n = 27; BHB n = 26; GABOB n = 23; Retigabine n = 15). **e** Effects of small molecules as indicated (100 μM) on [3H]Myo-inositol uptake by SMIT1 co-expressed with KCNQ2-W236L/KCNQ3-W265L (Q2WL/Q3WL) subunits in Xenopus oocytes ([3H]Myo-inositol n = 14; GABA n = 25; BHB n = 18; GABOB n = 28).

KCNQ2/3 activation (Fig. 2h). SMIT1 modulated the effects of BHB on KCNQ2/3, but differently to effects for GABA. Thus, SMIT1 increased the maximal efficacy of BHB 3-fold with respect to KCNQ2/3, yielding an maximal 8-fold increase in current at −60 mV (Fig. 2i) but reduced the BHB potency 60-fold as quantified by fold increase at −60 mV (KCNQ2/3–SMIT1, BHB $EC_{50} = 16.5 \pm 0.46$ μM, versus $0.26 \pm 0.07$ μM for KCNQ2/3 without SMIT1) (Fig. 2i) or 13-fold as quantified by $\Delta V_{0.5activation}$ (KCNQ2/3–SMIT1, BHB $EC_{50} = 10.2 \pm 0.31$ μM, versus $0.79 \pm 0.30$ μM for KCNQ2/3 without SMIT1) (Fig. 2j).

GABOB is a high-affinity partial agonist of KCNQ2/3 channels[8]. GABOB effects on KCNQ2/3–SMIT1 voltage dependence and activation rate were minimal (Fig. 2k–n). SMIT1 appears to reduce KCNQ2/3 sensitivity, by threefold as quantified by $\Delta V_{0.5activation}$ (KCNQ2/3–SMIT1 GABOB $EC_{50} = 304 \pm 100$ nM, versus $89.1 \pm 10$ nM for KCNQ2/3 alone) (Fig. 2o). The capacity of SMIT1 to tune KCNQ2/3 sensitivity to GABA and related metabolites suggests that SMIT1 impinges on the KCNQ2/3 binding site for these molecules.

**GABA and SMIT1 similarly alter KCNQ2/3 ion selectivity.** We previously found that SMIT1 binds to the KCNQ2 pore module

and decreases KCNQ2/3K+ selectivity, producing a relative increase in Na+ and Cs+ permeability—evidence of an effect of SMIT1 binding on KCNQ2/3 pore conformation[13]. Here, we found that GABA, which binds to a conserved tryptophan in S5 within the pore module of KCNQ2–5 (e.g., W265 in KCNQ3), also increases KCNQ2/3 Na+ and Cs+ permeability relative to that of K+ (Fig. 3a–c). The results suggest that there are some common features to GABA and SMIT1 effects on KCNQ2/3 pore conformation. We next tested the effects of GABA on homomeric KCNQs (Fig. 3d–l). We previously found that while GABA binds to the S5 tryptophan of both KCNQ2 (W236) and KCNQ3 (W265), GABA activates homomeric KCNQ3, but not homomeric KCNQ2[8]. Here, we found that GABA has no effect on the ion selectivity of homomeric KCNQ2 (Fig. 3d–f). In contrast, GABA increased the Na+ and Cs+ permeability of homomeric KCNQ3*—an expression-optimized KCNQ3 mutant (A315T), previously found to give robust KCNQ3 currents without altering other fundamental channel properties[19] (Fig. 3g–i). As an additional control we examined KCNQ4, which also binds GABA but is not activated by it[8]. GABA also had no effect on KCNQ4 ion selectivity (Fig. 3j–l). Thus, the current-potentiating effects of GABA occur concomitant to a shift in ion selectivity (as we

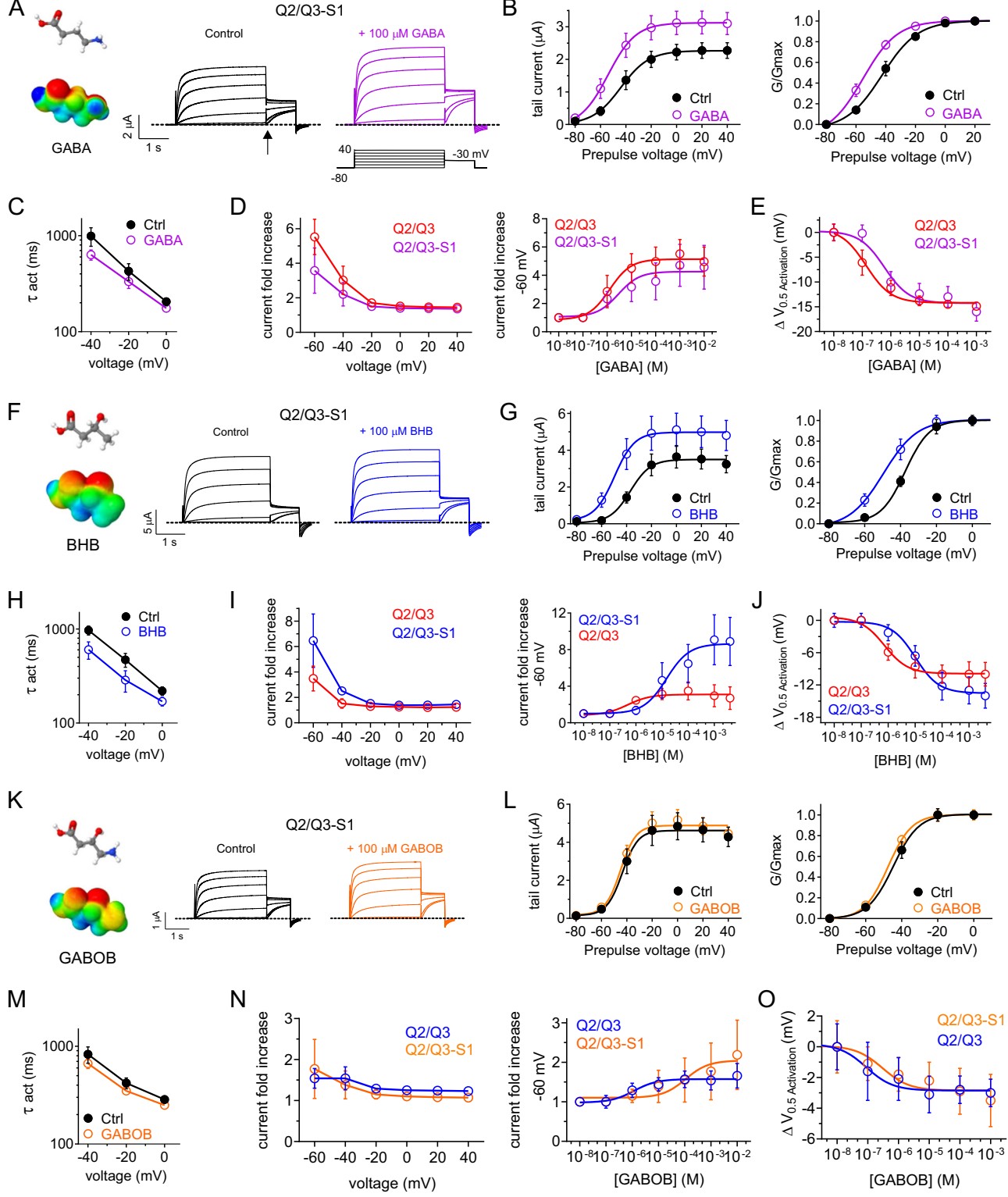

observe for KCNQ3 homomers and KCNQ2/3 heteromers); the selectivity shift does not occur in KCNQs that are not activated by GABA, even when GABA can bind to them (homomeric KCNQ2 and KCNQ4). These data suggest that the ion selectivity shift signals a conformational shift in the pore that occurs upon activation by GABA.

**KCNQ3 is structurally primed to be the GABA sensor for SMIT1.** The effects of SMIT1 on homomeric KCNQ3 channels were not previously reported. Here, we found that SMIT1 positively shifts the voltage dependence of KCNQ3* (Fig. 4a–d), slows its activation (Fig. 4e) and depolarizes KCNQ3*-expressing oocytes (Fig. 4f), i.e., SMIT1 impedes voltage-dependent KCNQ3* activation. This is the opposite of SMIT1 effects on KCNQ2 and KCNQ2/3, as we previously reported[13]. Comparing next the effects of SMIT1 on homomeric KCNQ2 and KCNQ3* relative ion permeability, we found that SMIT1 greatly increases the relative $Na^+$ and $Cs^+$ permeation compared to $K^+$ of KCNQ2 (Fig. 4g–i), but not

**Fig. 2 SMIT1 alters effects of GABA on KCNQ2/KCNQ3 channels.** All error bars indicate SEM. **a** Mean traces for KCNQ2/KCNQ3 (Q2/Q3) co-expressed with SMIT1 (S1) in the absence (Control) or presence of GABA (100 μM) ($n = 5$). Voltage protocol for this and all similar recordings in this study, inset. Structure and electrostatic surface potential heatmap plots shown for GABA. *Red*, negative charge, *blue*, positive charge for all heatmaps in this article. **b** Mean tail current (at arrow in (**a**)) and normalized tail current ($G/Gmax$) versus prepulse voltage for KCNQ2/KCNQ3–SMIT1 traces as in (**a**) ($n = 5$). **c** GABA effects on KCNQ2/KCNQ3–SMIT1 activation time constant $= \tau_{act}$ ($n = 5$). **d** Voltage dependence and dose response (at −60 mV) for GABA effects on KCNQ2/KCNQ3 (red; $n = 5$) and KCNQ2/KCNQ3–SMIT1 (magenta; $n = 6$) current. **e** GABA dose response for effects on KCNQ2/KCNQ3 (red; $n = 5$) and KCNQ2/KCNQ3–SMIT1 (magenta; $n = 6$) activity (shift in midpoint voltage of activation). **f** Mean traces for KCNQ2/KCNQ3 (Q2/Q3) co-expressed with SMIT1 (S1) in the absence (control) or presence of BHB (100 μM) ($n = 5$). **g** Mean and normalized tail current ($G/Gmax$) versus prepulse voltage for KCNQ2/KCNQ3–SMIT1 traces as in (**f**) ($n = 5$). **h** BHB effects on KCNQ2/KCNQ3–SMIT1 activation rate ($n = 5$). **i** Voltage dependence and dose response (at −60 mV) for BHB effects on KCNQ2/KCNQ3 (red) and KCNQ2/KCNQ3–SMIT1 (blue) ($n = 5$). **j** BHB dose response for effects on KCNQ2/KCNQ3 (red) and KCNQ2/KCNQ3–SMIT1 (blue) activity (shift in midpoint voltage of activation) ($n = 5$). **k** Mean traces for KCNQ2/KCNQ3 (Q2/Q3) co-expressed with SMIT1 (S1) in the absence (control) or presence of GABOB (100 μM) ($n = 5$). **l** Mean and normalized tail current ($G/Gmax$) versus prepulse voltage for KCNQ2/KCNQ3–SMIT1 traces as in (**k**) ($n = 5$). **m** GABOB effects on KCNQ2/KCNQ3–SMIT1 activation rate ($n = 5$). **n** Voltage dependence and dose response (at −60 mV) for GABOB effects on KCNQ2/KCNQ3 (blue; $n = 5$) and KCNQ2/KCNQ3–SMIT1 (orange; $n = 6$). **o** GABOB dose response for effects on KCNQ2/KCNQ3 (blue; $n = 5$) and KCNQ2/KCNQ3–SMIT1 (orange; $n = 6$) activity (shift in midpoint voltage of activation).

KCNQ3* (Fig. 4j–l). This is again consistent with the increased Na$^+$ and Cs$^+$ permeability being the signature of an alternate pore conformation that arises when the voltage dependence of KCNQ channel activation is negative shifted, either by GABA or by SMIT1 (Fig. 3 and ref. [13]). This suggests that while GABA activates and alters pore properties of KCNQ2/3 channels via KCNQ3, SMIT1 activates and alters pore properties of KCNQ2/3 via KCNQ2. Furthermore, the results suggest that by maintaining its baseline pore conformation despite interaction with SMIT1, KCNQ3 (but not KCNQ2) is structurally primed to respond to GABA and communicate the effects of GABA binding to SMIT1 to regulate SMIT1 function. Consistent with this hypothesis, KCNQ3*–SMIT1 but not KCNQ2–SMIT1 was responsive to GABA, with the functional output being an approximately twofold reduction in SMIT1 *myo*-inositol uptake (Fig. 4m, n), similar to effects of GABA on KCNQ2/3–SMIT1 complexes (Fig. 1d).

**SMIT1 permits KCNQs to distinguish BHB from GABA and GABOB.** We previously showed that, as for GABA, BHB activates homomeric KCNQ3* but not KCNQ2[8]. Strikingly, here we found that while KCNQ3* but not KCNQ2 can communicate the GABA binding event to co-expressed SMIT1 to alter its activity, for BHB the isoform selectivity is reversed. Thus, BHB inhibits *myo*-inositol transport activity of SMIT1 when co-expressed with KCNQ2, but not KCNQ3*. In contrast, for GABOB, as for GABA, KCNQ3* but not KCNQ2 conferred sensitivity to SMIT1 (Fig. 4m, n). Thus, SMIT1 enables KCNQs to distinguish BHB from the structurally highly similar (Fig. 2) GABA and GABOB.

**An S4–5 arginine is required for SMIT1–KCNQ2/3 communication.** We previously found that GABA binds to the S5-located KCNQ2–W236 and KCNQ3–W265 (Fig. 5a, b), and that activation of KCNQ2/3 channels requires KCNQ3–W265 (KCNQ2 binds GABA but is not activated by it)[8]. We showed earlier that complexes formed by SMIT1 and heteromeric mutant W236L–KCNQ2/W265L–KCNQ3 channels do not respond to GABA (Fig. 1e) which is consistent with the role of the S5 W residues in GABA binding. However, W236L–KCNQ2/W265L–KCNQ3 channels remain sensitive to SMIT1, which negative shifts their midpoint voltage dependence of activation (Fig. 5c–f) and hyperpolarizes W236L–KCNQ2/ W265L–KCNQ3-expressing oocytes (Fig. 5g) as we previously found for wild-type KCNQ2/3 channels[13].

We also recently found that some plant metabolites can hyperpolarize KCNQ channel activation voltage dependence by binding to an arginine (e.g., R242 in KCNQ3) at the foot of S4, abutting the S4–5 linker (Fig. 5b)[20–23]. More recently, we found

that this arginine also influences GABA binding[23]. Here, the activity of KCNQ2/3 channels with the S4–5 arginine mutated to alanine (KCNQ2–R213A/KCNQ3–R242A) was strongly inhibited by SMIT1 (Fig. 5h, i), in contrast to effects for wild-type[13] or WL/ WL (Fig. 5c–g) KCNQ2/3. While the low current magnitude of the resultant channels precluded accurate quantification of voltage dependence, SMIT1 co-expression positive shifted the resting membrane potential of KCNQ2–R213A/KCNQ3–R242A channels (Fig. 5j), the opposite of effects on wild-type[13] or WL/ WL (Fig. 5c–g) KCNQ2/3. Thus, we conclude that GABA binding to the S5 W causes a conformational shift in KCNQ2/3 channels that is communicated to co-assembled SMIT1 via residues other than the S5 W, likely including R213/R242. In accord with this hypothesis, KCNQ2–R213A/KCNQ3–R242A channels were unable to modulate either SMIT1 *myo*-inositol transport, or the regulation of this process by GABA (Fig. 5k).

## Discussion

An increasing number of potassium channel-transporter ("chansporter") complexes are being uncovered. Thus far, the KCNQ potassium channel family appears particularly influential in this area, but whether this stems from an inherent proclivity of KCNQs to participate in chansporter complexes, or an investigational bias arising from the early KCNQ studies in this area is not yet known. KCNQ1 physically interacts with SMIT1 in the choroid plexus, in tripartite complexes with KCNE2, and in vitro is thought to also form complexes with the structurally related *myo*-inositol transporter, SMIT2[13,24]. KCNQ2/3 complexes are now known to interact with SMIT1 and SMIT2[13,14], and DAT and GLT1 (sodium-coupled dopamine and glutamate transporters, respectively)[25,26] in the brain and/or peripheral nervous system.

The single *Drosophila* KCNQ gene product, dKCNQ, interacts with *cupcake* (*dSLC5A11*), a non-transporting orthologue of SMIT2 that probably acts as a glucose sensor to help control feeding behavior via its effect on dKCNQ activity[27]. Similar to what we found for human SMIT2 with KCNQ1[24], dSLC5A11 inhibits dKCNQ activity; increased *cupcake* expression in the vinegar fly brain inhibits neuronal dKCNQ activity. This promotes feeding and other hunger-driven activities[27].

In complexes with the DAT and GLT1 transporters, KCNQ2/3 channels are thought to counteract the depolarizing force of sodium co-transport into neurons, facilitating optimal neurotransmitter transport through DAT and GLT1 while also preventing excessive membrane depolarization. As previously found for KCNQ–SMIT chansporter complexes[24], inhibition of the co-assembled KCNQ2/3 channels using the small molecule XE991 removed the augmenting effects of KCNQ2/3 on DAT and GLT1 transport activity. Complexes formed from DAT and GLT1 with KCNQ2/3 may arise in

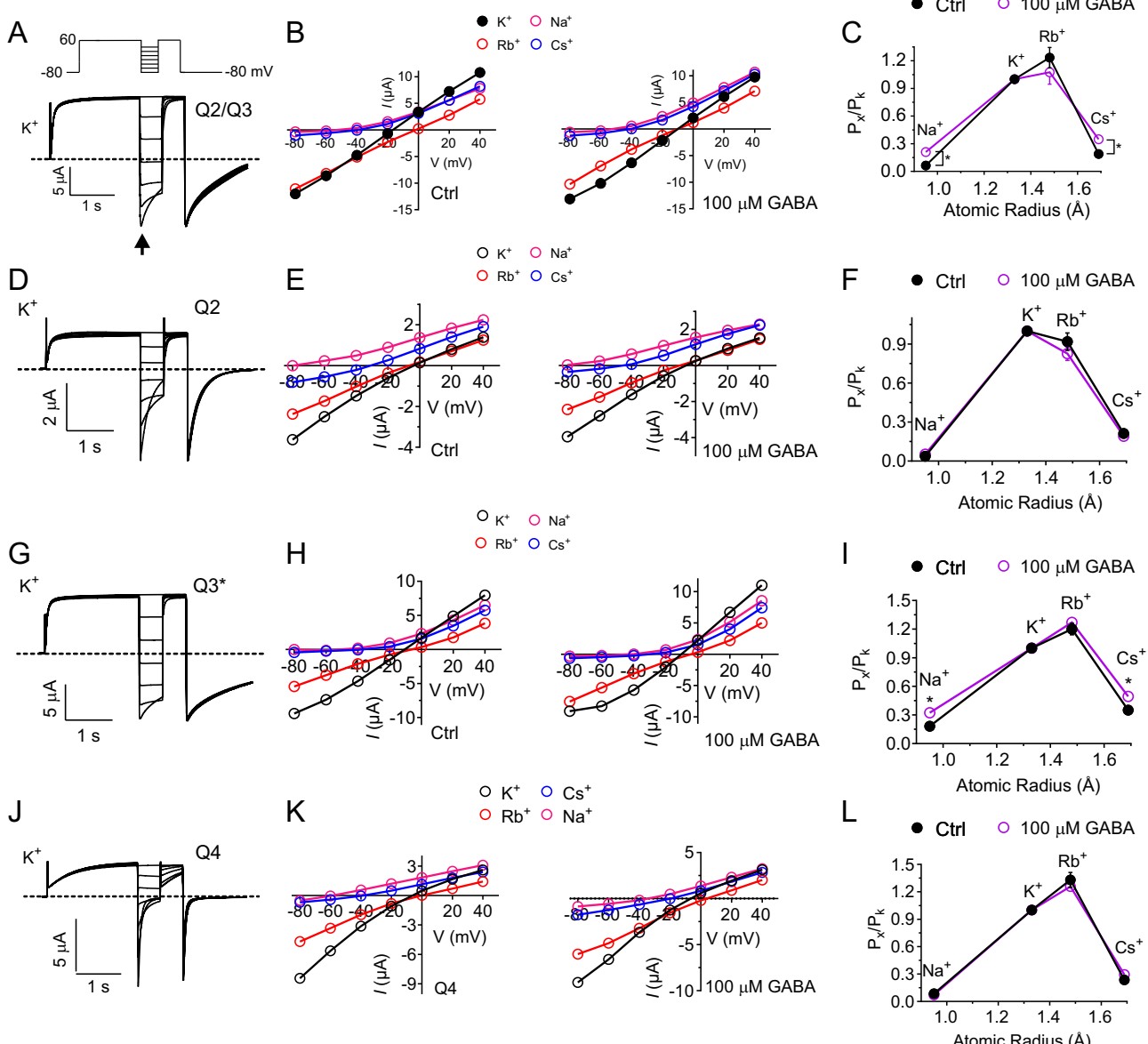

**Fig. 3 GABA increases relative sodium and cesium permeability of KCNQ channels that it activates.** All error bars indicate SEM; n values indicate biologically independent experiments. **a** Representative traces from recordings of *Xenopus laevis* oocytes injected with cRNA encoding KCNQ2/3 in 100 mM $K^+$ (voltage protocol inset). **b** Mean current–voltage relationship for KCNQ2/3 in 100 mM $K^+$ (black), $Cs^+$ (blue), $Rb^+$ (red) and $Na^+$ (magenta; $n = 5$), in the absence or presence of GABA (100 µM; $n = 6$) as indicated, values quantified from arrow in panel (**a**). **c** Estimated mean ion permeability relative to that of $K^+$ versus ionic radius (Pauling) through KCNQ2/Q3 in the absence (black; $n = 5$) and presence (purple; $n = 6$) of GABA (100 µM). *$p < 0.05$. **d** Representative traces from recordings of oocytes injected with cRNA encoding KCNQ2 in 100 mM $K^+$. **e** Mean current–voltage relationship for KCNQ2 in 100 mM $K^+$ (black), $Cs^+$ (blue), $Rb^+$ (red), and $Na^+$ (magenta), in the absence or presence of GABA (100 µM) as indicated; $n = 5$. **f** Estimated mean ion permeability relative to that of $K^+$ versus ionic radius (Pauling) through KCNQ2 in the absence (black) and presence (purple) of GABA (100 µM); $n = 5$. **g** Representative traces from recordings of oocytes injected with cRNA encoding KCNQ3* in 100 mM $K^+$. **h** Mean current–voltage relationship for KCNQ3* in 100 mM $K^+$ (black), $Cs^+$ (blue), $Rb^+$ (red) and $Na^+$ (magenta), in the absence or presence of GABA (100 µM) as indicated; $n = 4$. **i** Estimated mean ion permeability relative to that of $K^+$ versus ionic radius (Pauling) for $Na^+$, $Rb^+$, and $Cs^+$ through KCNQ3* in the absence (black) and presence (purple) of GABA (100 µM); $n = 4$. *$p < 0.05$. **j** Representative traces from recordings of oocytes injected with cRNA encoding KCNQ4 in 100 mM $K^+$. **k** Mean current–voltage relationship for KCNQ4 in 100 mM $K^+$ (black), $Cs^+$ (blue), $Rb^+$ (red), and $Na^+$ (magenta), in the absence or presence of GABA (100 µM) as indicated; $n = 4$. **l** Estimated mean ion permeability relative to that of $K^+$ versus ionic radius (Pauling) through KCNQ4 in the absence (black) and presence (purple) of GABA (100 µM); $n = 4$.

axons where the restrictive confines could necessitate complex formation to facilitate efficient channel-transporter cross-talk[25] as we also proposed for KCNQ2/3–SMIT1/2 complexes[14].

Outside the KCNQ family, the $Ca^{2+}$-activated $K+$ channel BK was found to form complexes with the GABA transporter 3, GAT3—another sodium-coupled transporter (encoded by *SLC6A11*)—in mouse brain lysates[28]. Similarly, KCNA2 (Kv1.2) interacts with the LAT1 (*SLC7A5*) neutral amino acid transporter and the two co-localize in mouse neurons. LAT1 regulates KCNA2 gating and voltage dependence, and the proteins each

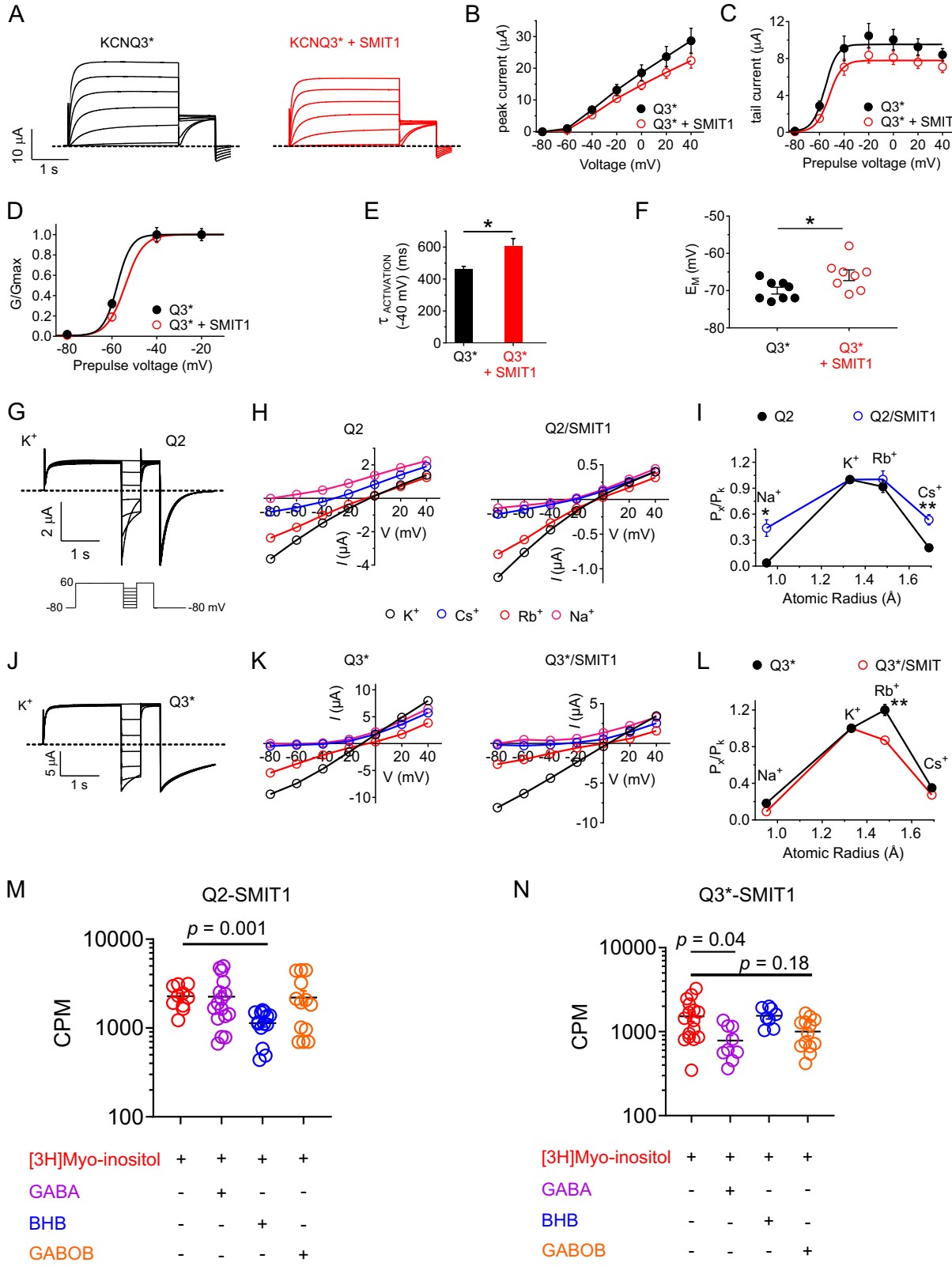

alter the effects in one another of gene variants linked to developmental delay and epilepsy[29].

In the present study we provide evidence that KCNQ channels in chansporter complexes can act as a chemosensor for the transporter, enabling the latter to respond to stimuli it does not typically respond to, e.g., GABA, BHB, GABOB, retigabine. Reciprocally, the transporter has the capability to tune the

response of the channel to those same stimuli, at least in the case of SMIT1 altering how KCNQ channels sense GABA and related metabolites.

We demonstrate that KCNQ2/3 mediates GABA regulation of SMIT1 activity both in vitro and in ex vivo DRG. We do not yet know the physiological role of this regulation, but it could potentially represent a form of negative feedback regulation.

**Fig. 4 SMIT1–KCNQ complexes KCNQ-isoform dependently distinguish between GABA and related metabolites.** All error bars indicate SEM. **a** Mean traces showing currents generated by KCNQ3* alone or with SMIT1 ($n = 8$). **b** Mean peak currents for KCNQ3* alone (black) or with SMIT1 (red) quantified from traces as in (**a**) ($n = 8$). **c** Mean peak tail currents for KCNQ3* alone (black) or with SMIT1 (red) quantified from traces as in (**a**) ($n = 8$). **d** Mean G/Gmax relationship for KCNQ3* alone (black) or with SMIT1 (red) quantified from traces as in (**a**) ($n = 8$). **e** Effects of SMIT1 on KCNQ3* activation rate quantified as a single exponential fit, time constant = $\tau_{act}$ ($n = 8$). *$p < 0.05$. **f** Mean $E_M$ of unclamped oocytes expressing KCNQ3* alone (black) or with SMIT1 (red) ($n = 8$). *$p < 0.05$. **g** Representative traces from KCNQ2 (Q2) in 100 mM K$^+$ (voltage protocol inset). **h** Mean current–voltage relationship for Q2 in 100 mM K$^+$ (black), Cs$^+$ (blue), Rb$^+$ (red), and Na$^+$ (magenta), in the absence or presence of SMIT1 as indicated; quantified from traces as in (**g**); $n = 5$. **i** Estimated mean permeability relative to that of K$^+$ versus ionic radius (Pauling) through Q2 in the absence (black) and presence (blue) of SMIT1; $n = 5$. *$p < 0.05$; **$p < 0.01$. **j** Representative traces from KCNQ3* (Q3*) in 100 mM K$^+$ (voltage protocol as in (**g**)). **k** Mean current–voltage relationship for Q3* in 100 mM K$^+$ (black), Cs$^+$ (blue), Rb$^+$ (red), and Na$^+$ (magenta), with/without SMIT1 as indicated; quantified from traces as in (**j**); $n = 4$. **l** Estimated mean permeability relative to that of K$^+$ versus ionic radius (Pauling) through Q3* with (red) or without (black) SMIT1; $n = 4$. **$p = 0.006$. **m** [$^3$H]$Myo$-inositol uptake (30 min) for oocytes expressing KCNQ2 and SMIT1 in the absence or presence of neurotransmitters/metabolites indicated (100 μM) ([$^3$H]$Myo$-inositol $n = 10$; GABA $n = 16$; BHB $n = 17$; GABOB $n = 13$). **n** [$^3$H]$Myo$-inositol uptake (30 min) for oocytes expressing KCNQ3* and SMIT1 with/without neurotransmitters/metabolites indicated (100 μM) ([$^3$H]$Myo$-inositol $n = 17$; GABA $n = 10$; BHB $n = 8$; GABOB $n = 16$).

In this model, increased local GABA concentration subdues neuronal firing by activating KCNQ2/3, but this in turn inhibits co-assembled SMIT1, reducing local [PIP$_2$] and eventually leading to KCNQ2/3 inhibition, increasing excitability once again (see Fig. 6 for schematic).

It is important to mention that GABA binding to KCNQ2/3 is able to overcome inhibition of KCNQ2/3 channels by PIP$_2$ depletion and/or muscarinic receptor activation[8], therefore the potential excitatory effects of PIP$_2$ depletion would be predicted to come into effect only after GABA concentrations had sufficiently subsided. This combination of effects might be required for timed waves of excitation and inhibition, or conversely even to dampen such oscillations, depending on the magnitude and timing of GABA and PIP$_2$ concentration changes. Overlaid upon this, it was previously discovered that KCNQ2/3 channels themselves regulate GABA release[10,11], pointing to further possible layers of feedback regulation.

SMIT1, by apparently impacting the KCNQ-binding site for GABA and related small molecules, can tune KCNQ responses to these important neuronal signaling moieties. Perhaps most surprisingly, SMIT1 switched the KCNQ isoform selectivity of BHB such that its binding was communicated to SMIT1 not by KCNQ3 but by KCNQ2 (Fig. 5). We had previously found that while KCNQ2–5 (but not KCNQ1 because it lacks the crucial S5 tryptophan) also bind GABA, only KCNQ3 and KCNQ5 respond to GABA[8]. Similarly, BHB activates KCNQ2/3 heteromers and KCNQ3* homomers, but not KCNQ2 homomers[30]. As we also observed for GABA, when in complexes with KCNQ2/3 heteromers, SMIT1 is inhibited by BHB (Fig. 1). Surprisingly, and opposite to GABA and GABOB, SMIT1 was insensitive to BHB when in complexes with KCNQ3*, but was BHB-sensitive when coupled with KCNQ2 (Fig. 4). SMIT1 therefore endows KCNQs with an additional capacity to distinguish between GABA and close structural orthologues that are both present endogenously in the nervous system (BHB) and altered in a therapeutically/pathologically relevant manner: BHB is the first ketone body produced during fasting or ketogenic diets, and also in diabetic ketoacidosis. Occipital lobe β-hydroxybutyrate concentrations rise in children from around 50 μM at baseline to approximately 1 mM after 72 h of fasting[31]. SMIT1 both alters KCNQ selectivity for BHB, and also amplifies the effects of clinically relevant concentrations of BHB on KCNQ2/3, suggesting we should consider KCNQ2/3–SMIT1 complexes when evaluating the mechanisms underlying the therapeutic action of the ketogenic diet and fasting in epilepsy.

We also show for the first time, to our knowledge, in this study that GABA induces a distinct pore conformation, characterized by increased relative permeability of Na$^+$ and Cs$^+$ relative to K$^+$.

This new conformation only arises when GABA activates KCNQs; it does not occur in homomeric KCNQ2 or KCNQ4 (Fig. 3), which bind GABA but are not activated by it[8]. These data support an hypothesis in which rather than GABA simply shifting the voltage dependence of KCNQ2/3 by favoring voltage sensor movement more readily at a given membrane potential, GABA actually induces a distinct pore conformation/activation state with different voltage dependence than the GABA-free channel.

In summary, we demonstrate channel-transporter co-assembly brings new properties to one or both entities within the complex, and most strikingly, the channels act as chemosensors for their partner transporters. This, to our knowledge, novel role for potassium channels gives further clues to the evolutionary forces that have brought channels and transporters together. It also has additional ramifications for therapeutic targeting of either protein type, beyond that mentioned above for BHB. For example, we demonstrate that the anticonvulsant retigabine inhibits SMIT1 activity (via co-assembled KCNQ2/3) (Fig. 1). Retigabine was removed from the clinic in 2017 because of side effects of unknown mechanistic origin, including discoloration of the skin, sclera and nails[32]. While the discoloration effects are unlikely to occur because of SMIT1 modulation, other SMIT1-dependent effects of retigabine and related compounds may either act therapeutically or work against its intended activity, suggesting that SMIT1 activity assays might be included when screening for new KCNQ-opening anticonvulsants such that the possible drawbacks and advantages can be fully investigated.

## Methods

**cRNA preparation and and *X. laevis* oocyte injection.** cRNA transcripts encoding human KCNQ2, KCNQ3, KCNQ4, or SMIT1 were generated as before[8] by in vitro transcription using mMessage mMachine kits (Thermo Fisher Scientific), after cDNA linearization. cRNA was quantified by spectrophotometry. Mutant cDNAs were generated by site-directed mutagenesis using a QuikChange kit according to manufacturer's protocol (Stratagene, San Diego, CA) and the corresponding cRNAs were prepared as above. We injected defolliculated stage V and VI *X. laevis* oocytes (Ecocyte Bioscience, Austin, TX) with Kv channel α subunit cRNAs (5–20 ng) and/or SMIT1 cRNA (10–20 ng). Oocytes were incubated at 16 °C in Barth's saline solution (Ecocyte) containing penicillin and streptomycin, with daily washing, for 3–5 days prior to TEVC recording and/or radioligand uptake studies.

**Radioligand uptake studies.** For DRG studies, we conducted radiolabeled uptake assays using *myo*-[2-$^3$H(N)]inositol (American Radiolabeled Chemicals Inc.) on whole, freshly isolated postnatal day 8 rat DRG purchased from BrainBits (Springfield, Illinois). DRG were removed from the Hibernate AB storage solution (BrainBits) and placed into NbActiv4 medium (BrainBits) containing 25 ng/ml of nerve growth factor and incubated overnight at 37 °C. Isolated DRG were cut into 5 mm pieces and placed in 1 ml Eppendorf tubes containing NbActiv4 medium (200 μl per tube). Depending on experimental conditions the medium contained *myo*-[2-$^3$H(N)]inositol (3 μCi/ml) alone or with 100 μM GABA (Sigma). After 30 min at room temperature, DRG were washed two to three times in 3 ml of fresh

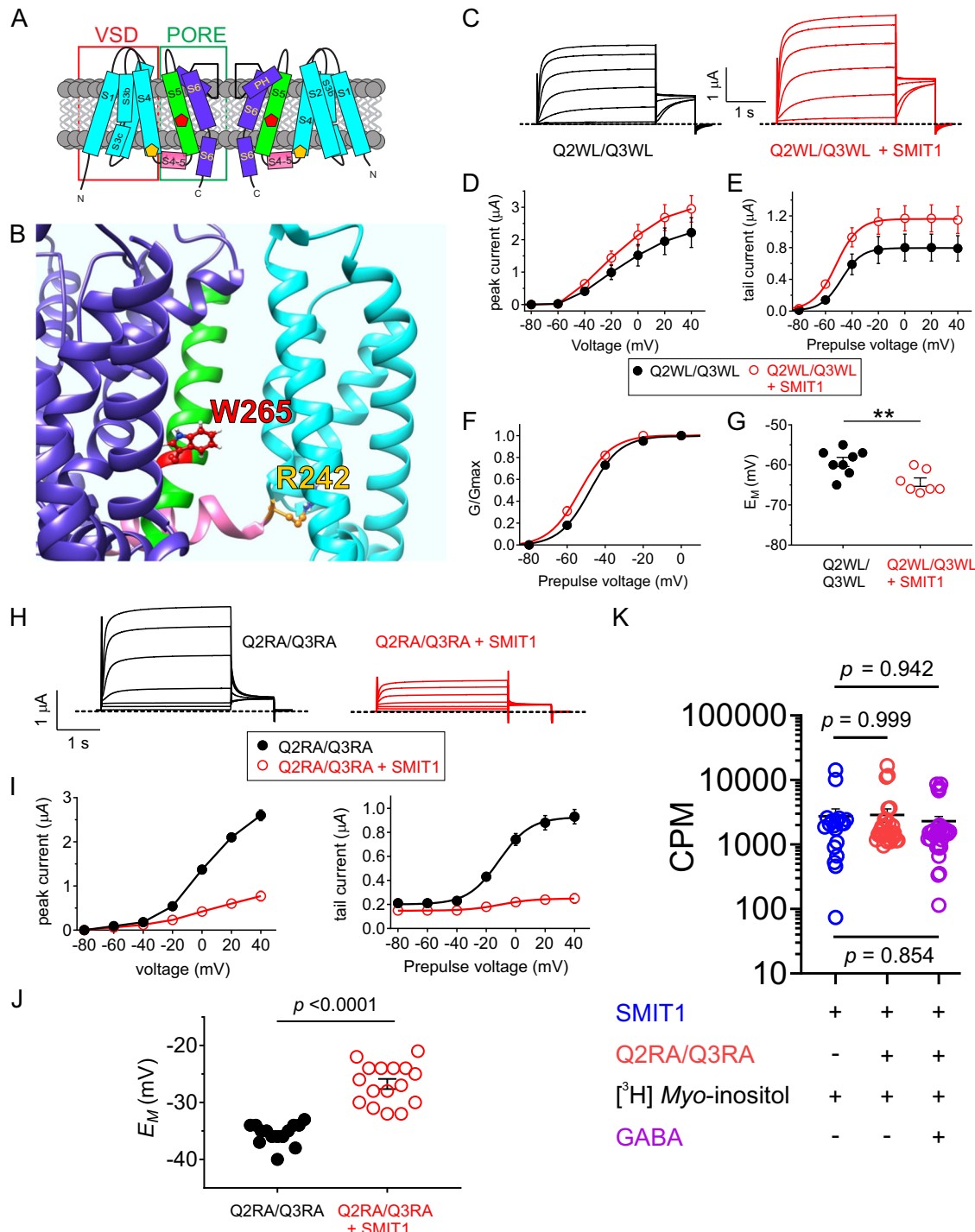

**Fig. 5 KCNQ2/3 channels require an arginine at the foot of the voltage sensor for communicating GABA binding to SMIT1.** All error bars indicate SEM. **a** Topology of KCNQ3 (two of four subunits shown) showing positions of W265 (red) and R242 (yellow) and segment numbering. VSD voltage-sensing domain. **b** Structural model of KCNQ3 showing positions of W265 and R242. Color coding as in (**a**). **c** Mean traces showing currents generated by KCNQ2–W236L/KCNQ3–W265L (Q2WL/Q3WL) ($n = 8$) alone or with SMIT1 ($n = 7$) in *Xenopus* oocytes. **d** Mean peak currents for KCNQ2–W236L/KCNQ3–W265L (Q2WL/Q3WL) alone (black; $n = 8$) or with SMIT1 (red; $n = 7$) quantified from traces as in (**c**). **e** Mean peak tail currents for KCNQ2–W236L/KCNQ3–W265L (Q2WL/Q3WL) alone (black; $n = 8$) or with SMIT1 (red; $n = 7$) quantified from traces as in (**c**). **f** Mean $G/G$max relationship for KCNQ2–W236L/KCNQ3–W265L (Q2WL/Q3WL) alone (black; $n = 8$) or with SMIT1 (red; $n = 7$) quantified from traces as in (**c**). **g** Mean $E_M$ of unclamped *Xenopus* oocytes expressing KCNQ2–W236L/KCNQ3–W265L (Q2WL/Q3WL) alone (black) ($n = 8$) or with SMIT1 (red) ($n = 7$). **p < 0.01. **h** Mean traces showing currents generated by KCNQ2–R213A/KCNQ3–R242A (Q2RA/Q3RA) alone ($n = 14$) or with SMIT1 ($n = 16$) in *Xenopus* oocytes. **i** Mean peak prepulse and peak tail currents for KCNQ2–R213A/KCNQ3–R242A (Q2RA/Q3RA) alone (black; $n = 14$) or with SMIT1 ($n = 16$) quantified from traces as in panel (**h**). **j** Mean $E_M$ of unclamped *Xenopus* oocytes expressing KCNQ2–R213A/KCNQ3–R242A (Q2RA/Q3RA) alone (black) ($n = 14$) or with SMIT1 (red) ($n = 16$). **k** [³H]*Myo*-inositol uptake (30 min) for *Xenopus* oocytes expressing subunits indicated in the absence or presence of GABA (100 μM) as indicated (SMIT1 alone, $n = 19$; Q2RA/Q3RA-S1, $n = 29$; Q2RA/Q3RA-S1 + GABA, $n = 32$).

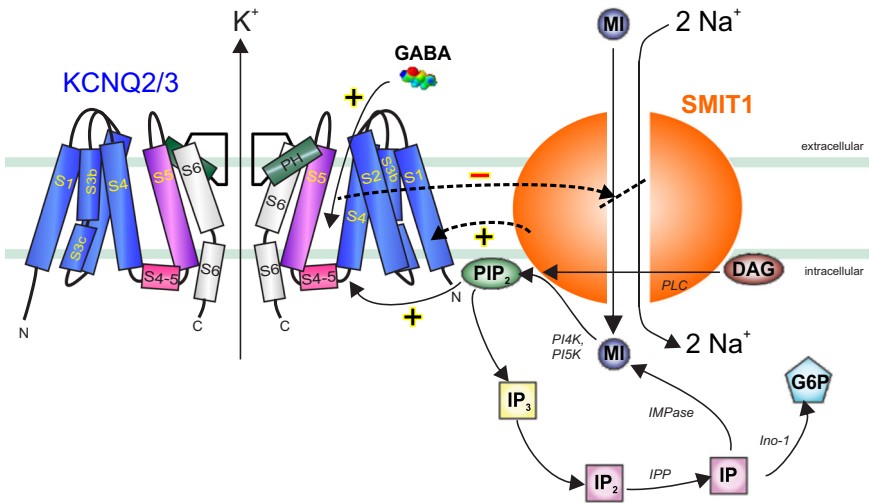

**Fig. 6 Model of signaling in KCNQ2/KCNQ3–SMIT1 complexes.** Schematic of KCNQ2/3–SMIT1 interaction showing functional effects of physical interaction and modulation by small molecules. DAG diacylglycerol, G6P glucose-6-phosphate, IMPase inositol monophosphatase, Ino-1 inositol-1 phosphate synthase, IP inositol phosphate, IPP inositol polyphosphatase 1-phosphatase, MI *myo*-inositol, PI4K phosphatidylinositol 4-kinase, PI5K phosphatidylinositol-5 kinase, PLC phospholipase C.

NbActiv4 medium containing 1 mM cold *myo*-inositol (Sigma), followed by 1–2 washes in 3 ml NbActiv4 medium. DRG were homogenized over a period of 6 h in 100 μl of 2% sodium dodecyl sulfate (SDS) in NbActiv4; each homogenized DRG section was then transferred to a scintillation vial (1 DRG section per vial) containing Ready Protein Plus scintillation fluid (Beckmann Coulter) (6 ml). Vials were capped, shaken vigorously, and then allowed to sit at room temperature for at least 30 min before scintillation counting in a Beckmann Coulter 6500. Each experiment was performed on two batches of DRG from different deliveries; each experiment always contained intra-batch controls and results presented are pooled data from both batches.

For oocyte studies, we conducted radiolabeled uptake assays using *myo*-[2–3H (N)]inositol (American Radiolabeled Chemicals Inc.) 5 days after oocyte cRNA injection. For each condition/expression group, oocytes were placed in a round-bottomed, 10-ml culture tube, washed, and resuspended in ND96 (200 μl per tube) containing *myo*-[2–3H(N)]inositol (3 μCi/ml) with or without GABA, glutamate, β-Hydroxybutyric acid (BHB), GABOB (Sigma), or retigabine (Tocris) at pH 7.5. After 30 min at room temperature, oocytes were washed 2–3 times in 3 ml of ND96 containing 1 mM cold *myo*-inositol (Sigma), followed by 1–2 washes in 3 ml of ND96. Oocytes were next individually placed in wells of a 96-well plate and lysed in 100 μl of 2% SDS in ND96; each lysed oocyte was then transferred to a scintillation vial (1 oocyte per vial) containing Ready Protein Plus scintillation fluid (Beckmann Coulter) (6 ml). Vials were capped, shaken vigorously, and then allowed to sit at room temperature for at least 30 min before scintillation counting in a Beckmann Coulter 6500. Each experiment was performed on several batches of oocytes from different deliveries; each experiment always contained intra-batch controls and results presented are pooled data from two or more batches.

**Two-electrode voltage clamp (TEVC).** We conducted TEVC recording at room temperature as before[8] with an OC-725C amplifier (Warner Instruments, Hamden, CT) and pClamp8 software (Molecular Devices, Sunnyvale, CA) 3–5 days after cRNA injection as described in section above. Oocytes were placed in a small-volume oocyte bath (Warner) and viewed with a dissection microscope. Chemicals were sourced from Sigma. Bath solution was (in mM): 96 NaCl, 4 KCl, 1 MgCl₂, 1 CaCl₂, 10 HEPES (pH 7.6). GABA and GABOB were stored at −80 °C as 1 M stocks in molecular grade H₂O and diluted to working concentrations on each experimental day. BHB was stored at 4 °C as a 480 mM stock in 100% ethanol and diluted to working concentrations each experimental day. All compounds were introduced to the recording bath via gravity perfusion at a constant flow of 1 ml per minute for 3 min prior to recording. Pipettes were of 1–2 MΩ resistance when filled with 3 M KCl.

Currents were recorded in response to pulses between −80 mV and +40 mV at 20 mV intervals, or a single pulse to +40 mV, from a holding potential of −80 mV, to yield current–voltage relationships, current magnitude, and for quantifying activation rate. TEVC data analysis was performed with Clampfit (Molecular Devices) and Graphpad Prism software (GraphPad, San Diego, CA, USA); values are stated as mean ± SEM. Normalized tail currents were plotted versus prepulse voltage and fitted with a single Boltzmann function

$$g = \frac{(A_1 - A_2)}{\left\{1 + \exp\left[\left(V_{1/2} - V\right)/Vs\right]\right\} y + A_2}, \qquad (1)$$

where $g$ is the normalized tail conductance, $A_1$ is the initial value at $-\infty$, $A_2$ is the final value at $+\infty$, $V_{1/2}$ is the half-maximal voltage of activation and $V_s$ the slope factor. Activation and deactivation kinetics were fitted with single exponential functions.

**Relative permeability calculations.** According to the Goldman–Hodgkin–Katz (GHK) voltage equation

$$E_{rev} = \frac{RT/F \ln\left(P_K[K^+]_O + P_{Na}[Na]_O + P_{Cl}[Cl]_i\right)}{\left(P_K[K^+]_i + P_{Na}[Na]_i + P_{Cl}[Cl]_o\right)}, \qquad (2)$$

where $E_{rev}$ is the absolute reversal potential and $P$ is permeability. This permits calculation of the relative permeability of each ion if concentrations on either side of the membrane are known. A modified version of this equation was used here to determine relative permeability of two ions in a system in which only the extracellular ion concentration was known. Thus, relative permeability of Rb⁺, Cs⁺, and Na⁺ compared to K⁺ ions was calculated for all channels by plotting the $I/V$ relationships for each channel with each extracellular ion (100 mM) and comparing them to that with 100 mM extracellular K⁺ ion to yield a change in reversal potential (DE$_{rev}$) for each ion compared to that of K⁺. Permeability ratios for each ion (X) compared to K⁺ were then calculated as

$$\Delta E_{rev} = E_{rev,X} - E_{rev,K} = \frac{RT}{zF} \ln\frac{P_X}{P_K}. \qquad (3)$$

Values were compared between channel types and statistical significance assessed using ANOVA.

**Chemical structures, in silico docking, and sequence analyses.** Chemical structures and electrostatic surface potentials (range: −0.1 to 0.1) were plotted using Jmol, an open-source Java viewer for chemical structures in 3D: http://jmol.org/. Illustrations of KCNQ structure were based on the *X. laevis* KCNQ1 cryoEM structure[33], which we altered to incorporate KCNQ3 residues known to be important for GABA and mallotoxin binding[8,20,21], followed by energy minimization using the GROMOS 43B1 force field[34], in DeepView[35]. Thus, *X. laevis* KCNQ1 amino acid sequence LIT*TL*YIGF was converted to LIT*AW*YIGF, the underlined W being W265 in human KCNQ3. In addition, *X. laevis* KCNQ1 sequence WWGVVTVTTIGYGD was converted to WWG*LITL*ATI-GYGD, the underlined L being Leu314 in human KCNQ3. Surrounding non-mutated sequences are shown to illustrate the otherwise high sequence identity in these stretches. Structures are visualized in UCSF Chimera[36].

**Statistics and reproducibility.** All values are expressed as mean ± SEM. One-way ANOVA was applied for all other tests; if multiple comparisons were performed, a post hoc Tukey's HSD test was performed following ANOVA. All *p* values were two-sided. Statistical significance was defined as *p* < 0.05.

**Reporting summary.** Further information on research design is available in the Nature Research Reporting Summary linked to this article.

## Data availability
The authors declare that all data supporting the findings of this study are available within the article.

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

## Acknowledgements
This study was supported by the US National Institutes of Health, National Institute of General Medical Sciences and National Institute of Neurological Disorders and Stroke (GM115189, GM130377, and NS107671 to G.W.A.). We are grateful to Lily Chen and Angele De Silva (University of California, Irvine) for generating mutant channel constructs. Molecular graphics and analyses were performed with UCSF Chimera, developed by the Resource for Biocomputing, Visualization, and Informatics at the University of California, San Francisco, with support from NIH P41-GM103311.

## Author contributions
R.W.M. performed all the electrophysiological experiments, transport studies and data analysis, prepared most of the figure panels, and edited the manuscript. G.W.A. conceived the study, generated docking figures and other models, wrote the manuscript and prepared the figures.

## Competing interests
The authors declare no competing interests.
