## [Peer Review File (redacted) · Communications Biology]

Reviewers' comments:

Reviewer #1 (Remarks to the Author):

The manuscript by Manville and Abbott is very interesting. The authors present extensive evidence supporting the notion that association of ion channels and transporters plays a functional role in regulating the effects of neurotransmitters and small molecules on channels (through transporters) or transporters (via the associated channels). Although the authors had previously demonstrated that ion channels can associate to solute transporters, they now provide new data unveiling new potential roles of these associations and, more importantly, about their regulation by GABA and its metabolites. This idea will hopefully raise unprecedented and daring questions in the field.

The manuscript is very well written. The text and figures flow easily. The experiments seem well performed and the data are solid. The data include experiments in whole dorsal root ganglia that are then compared to experiments where SMIT1-KCNQ2/3 are heterologously expressed in *Xenopus* oocytes. Other combinations of channel-transporters are also tested in oocytes, showing functional interaction and regulation by GABA.

I only have a few suggestions that may help to improve the manuscript.

[REDACTED]

2.-The scheme in Fig. 8 is very useful and important to follow the complex interactions described in the manuscript. However, in its present form is confusing, mainly because the arrows and lines depicting the interactions are not clearly seen. Whereas the use of the atomic structures is attractive, I do not seem to find that they add any value to this figure. This study does not unveil specific residues involved in the regulation, which would require the use of the structures. I would suggest to improve the figure by:

- a) using simpler cartoons so that the arrows and symbols describing the functional relationships are clearer. The pore region and other regions related to function, at the detail level of this study, can be easily pointed out in a cartoon.
- b) Should the line referring to MI transport be an arrow pointing towards the interior of the cell?
- c) Finally, in these new cartoons I would suggest to use the names of the proteins that are referred to in the text, i.e. SMIT1 and KCNQ2/KCNQ3 (instead of those of the structures used in the current version of Fig. 8).

3.- [REDACTED]

4.- [REDACTED]

5.- [REDACTED]

4.-typographical error: line 80, "toplogy".

Reviewer #2 (Remarks to the Author):

The authors previously reported the activation of KCNQ channel by GABA and also the co-localization and mutual functional regulation, including the ion selectivity change, between KCNQ and STIM1. In this paper the authors performed more detailed analyses focusing especially on (1) specificity of the effect of the ligands, i.e. GABA, BHB and GABOB; (2) specificity of the response of KCNQ subunit, i.e. KCNQ2/3, KCNQ2 and KCNQ3.

This paper contains many important and interesting data. However, it is very hard to accurately understand the whole story to the detail, and I felt stressful to read this manuscript. I think the readability should be thoroughly improved. Comments [2], [3] in the following are in relation to the readability.

[1]

There are some surprising or unexpected data which cannot be simply interpreted. The authors tried to explain appropriately, but they are not the only possible explanation. Thus, I should say that the unexpected findings are no more than descriptive in the present form. They are interesting, but cannot understand the mechanisms as a whole, and a confusion remains. Examples are as follows.
(1) Inhibition, but not activation, of KCNQ3 by SMIT1 in the absence of GABA (Fig. 5A, B, C, D), in contrast to KCNQ2/3
(2) Inhibition of Myo-IP transport by BHB, not in Q3, but in Q2.
And some more

[2]

The cases and analyses about which the authors intend to present could be very roughly described as follows.

Combinations of Cases 1 x Cases 2 x Cases 3 x Cases 4 x Parameters

Cases 1: Q2/Q3 or Q2 or Q3

Cases 2: wt or WL or RA

Cases 3: (-) SMIT1 or (+) SMIT1

Cases 4: (-) agonist or (+) GABA or (+) BHB or (+) GABOB

Parameters : gV shift or Iamp or t-act or EC50 or ion selectivity or myo-IP transport

Thus, there are so many cases and parameters. Some have already been published in ref 13 and 14, and not shown in this paper. Some of them, which I think important, are missing. Thus, it is hard to precisely understand what are known/ presented and what are unknown. I felt stress to read this manuscript. Examples of missing data are as follows.

(1) Effect of BHB on the ion selectivity of (Q2/Q3, Q2 alone, Q3 alone)

(2) Effect of SMIT1 on (Q2RA alone, Q3RA alone) x (Iamp, gV, ion selectivity)

And many more

[3]

I think the order of presentation is not necessarily straightforward. Examples are as follows.

- (1) Fig.3: subunit specificity, Fig. 4 subunit mutant, Fig. 5: subunit specificity
- (2) Ion selectivity analyses show up twice in Figs. 3, 5.
- (3) BHB, GABOB analyses show up three times in Figs. 1, 2, 5.
- (4) myo-IP transport analyses show up three times in Figs. 1, 4, 5.
- (5) The mutant depicted in Fig. 4 (W265L) is already used in Fig. 1

As there are many cases and parameters, I of course understand it is not easy to write in a straightforward way, but I had an impression at the initial reading that the story goes back and forth and could be re-arranged for better reading.

[4]

[REDACTED]

[5]

About the activation of KCNQ2/3 by SMIT1. Is the presence of SMIT1 molecule itself is important, or does the uptake of myo-IP also contribute? Some experimental data (e.g. comparison of the effect of SMIT1 on KCNQ2/3 channel, between (-) myoIP and (+) myoIP in the bath solution) are awaited.

[6]

About the inhibition of SMIT1 by KCNQ2/3. Is the presence of KCNQ2/3 molecule itself is important, or does the ionic current flow and/or change of E_m also contribute? What will happen if non-conducting pore mutant is used?

[7]

The discussion about the negative feedback regulation (page 13) is very interesting, but I am afraid it is over discussion at this stage with very limited evidences.

Reviewer #3 (Remarks to the Author):

Previous experiments carried out in the laboratory of the authors of this article had revealed the existence of a physical interaction between the SMIT1 myo-inositol transporter and the KCNQ2/3 potassium channel. In an observation that in principle did not seem related, the authors had determined that GABA was able to interact specifically with these potassium channels. Now the authors have observed that in DRG neurons, GABA modulates SMIT1 activity. The oocyte expression of the channel and the transporter seems to recapitulate the effect of the GABA observed in neurons. Using this experimental system, they have determined that the effect of GABA (and related metabolites) on SMT1 depends on the presence of the channel, which by binding GABA to its S5 domain is able to transmit the information to the attached transporter. In addition, this communication is bidirectional, since SMT1 alters the effect of GABA and related metabolites on the gating of potassium channels. Through directed mutagenesis they identify an arginine residue located in the channel voltage sensor as a mediator of this interaction.

This part of the article represents a solid piece of work in which the existence of a dialogue between both proteins is proven. In a second part, perhaps more debatable, the authors try to raise the general issue of the existence of a crosstalk between potassium channels and transporters by including two additional examples. More than the hypothesis itself, which is attractive, because

experimental support for the mechanisms of this crosstalk for these other proteins is weaker. [REDACTED]

[REDACTED]

[REDACTED] This reviewer has nothing against the experiments shown, but the characterization of the crosstalk is more superficial, and also the meaning of these interactions in a more physiological system is still uncertain.

We sincerely thank the reviewers for their constructive critiques. We have answered the comments with considerable rewriting of the manuscript, editing of the final figure, and additional clarifications (discussed point-by-point below).

Reviewers' comments:

Reviewer #1 (Remarks to the Author):

The manuscript by Manville and Abbott is very interesting. The authors present extensive evidence supporting the notion that association of ion channels and transporters plays a functional role in regulating the effects of neurotransmitters and small molecules on channels (through transporters) or transporters (via the associated channels). Although the authors had previously demonstrated that ion channels can associate to solute transporters, they now provide new data unveiling new potential roles of these associations and, more importantly, about their regulation by GABA and its metabolites. This idea will hopefully raise unprecedented and daring questions in the field.

The manuscript is very well written. The text and figures flow easily. The experiments seem well performed and the data are solid. The data include experiments in whole dorsal root ganglia that are then compared to experiments where SMIT1-KCNQ2/3 are heterologously expressed in *Xenopus* oocytes. Other combinations of channel-transporters are also tested in oocytes, showing functional interaction and regulation by GABA.

I only have a few suggestions that may help to improve the manuscript.

1.-In the final lines of the abstract, the three complexes studied in the manuscript are mentioned (KCNQ2/3-SMIT1, [REDACTED]). However, the introduction only refers to KCNQ2/3 and SMIT1. It would be good to introduce the other complexes that are going to be studied in the introduction, rather than much later in the manuscript.

>In response to editorial and reviewer comments we have removed the studies on GAT3 and KCNQ1, upon which we will conduct a more detailed study for the future.

2.-The scheme in Fig. 8 is very useful and important to follow the complex interactions described in the manuscript. However, in its present form is confusing, mainly because the arrows and lines depicting the interactions are not clearly seen. Whereas the use of the atomic structures is attractive, I do not seem to find that they add any value to this figure. This study does not unveil specific residues involved in the regulation, which would require the use of the structures. I would suggest to improve the figure by:

- a) using simpler cartoons so that the arrows and symbols describing the functional relationships are clearer. The pore region and other regions related to function, at the detail level of this study, can be easily pointed out in a cartoon.
- b) Should the line referring to MI transport be an arrow pointing towards the interior of the cell?

c) Finally, in these new cartoons I would suggest to use the names of the proteins that are referred to in the text, i.e. SMIT1 and KCNQ2/KCNQ3 (instead of those of the structures used in the current version of Fig. 8).

>We have made all these changes to improve clarity (now Figure 6).

[Redacted text block]

[Redacted text block]

[Redacted text block]

[Redacted text block]

[Redacted text block]

[Redacted text block]

4.-typographical error: line 80, “toplogy”.

>Corrected.

Reviewer #2 (Remarks to the Author):

The authors previously reported the activation of KCNQ channel by GABA and also the co-localization and mutual functional regulation, including the ion selectivity change, between KCNQ and STIM1.

In this paper the authors performed more detailed analyses focusing especially on (1) specificity of the effect of the ligands, i.e. GABA, BHB and GABOB; (2) specificity of the response of KCNQ subunit, i.e. KCNQ2/3, KCNQ2 and KCNQ3. They also studied the functional interactions between KCNQ1 and SMIT1, and also between BK and GAT3.

This paper contains many important and interesting data. However, it is very hard to accurately understand the whole story to the detail, and I felt stressful to read this manuscript. I think the readability should be thoroughly improved. Comments [2], [3] in the following are in relation to the readability.

[1]

There are some surprising or unexpected data which cannot be simply interpreted. The authors tried to explain appropriately, but they are not the only possible explanation. Thus, I should say that the unexpected findings are no more than descriptive in the present form. They are interesting, but cannot understand the mechanisms as a whole, and a confusion remains.

Examples are as follows.

(1) Inhibition, but not activation, of KCNQ3 by SMIT1 in the absence of GABA (Fig. 5A, B, C, D), in contrast to KCNQ2/3

(2) Inhibition of Myo-IP transport by BHB, not in Q3, but in Q2.

And some more

>We agree that some aspects lack a mechanistic basis at the molecular/atomic level, which would require high resolution structural analysis, which may be some years off for e.g., the differential effects of BHB and GABA with respect to KCNQ isoform selectivity. However, we consider it important to report these effects as they have potentially important physiological implications. Other aspects of the study do provide mechanistic insights as we delineate the specific residues involved and show that there is a shift in the pore conformation concomitant with activation that does not occur with GABA binding in the absence of activation.

[2]

The cases and analyses about which the authors intend to present could be very roughly described as follows.

Combinations of Cases 1 x Cases 2 x Cases 3 x Cases 4 x Parameters

Cases 1: Q2/Q3 or Q2 or Q3

Cases 2: wt or WL or RA

Cases 3: (-) SMIT1 or (+) SMIT1

Cases 4: (-) agonist or (+) GABA or (+) BHB or (+) GABOB

Parameters : gV shift or Iamp or t-act or EC50 or ion selectivity or myo-IP transport

Thus, there are so many cases and parameters. Some have already been published in ref 13 and 14, and not shown in this paper. Some of them, which I think important, are missing. Thus, it is hard to precisely understand what are known/ presented and what are unknown. I felt stress to read this manuscript. Examples of missing data are as follows.

- (1) Effect of BHB on the ion selectivity of (Q2/Q3, Q2 alone, Q3 alone)
 - (2) Effect of SMIT1 on (Q2RA alone, Q3RA alone) x (Iamp, gV, ion selectivity)
- And many more

>We have rearranged the manuscript to hopefully make it clearer to the reviewer. We are hesitant to add yet more data as this can make the manuscript unwieldy and cause further readability problems. In studies such as this there are also more and more combinations one can try but in the end we focused on the essential data that get the point across; there are already many different combinations studied.

[3]

I think the order of presentation is not necessarily straightforward. Examples are as follows.

- (1) Fig.3: subunit specificity, Fig. 4 subunit mutant, Fig. 5: subunit specificity
- (2) Ion selectivity analyses show up twice in Figs. 3, 5.
- (3) BHB, GABOB analyses show up three times in Figs. 1, 2, 5.
- (4) myo-IP transport analyses show up three times in Figs. 1, 4, 5.
- (5) The mutant depicted in Fig. 4 (W265L) is already used in Fig. 1

As there are many cases and parameters, I of course understand it is not easy to write in a straightforward way, but I had an impression at the initial reading that the story goes back and forth and could be re-arranged for better reading.

>We have tried various orders to best represent and rationalize the large quantity of results. In response to the reviewer's concern we have moved the data for the mutant channels to the end of the results so that the selectivity series studies are grouped together.

[REDACTED]

[4]

[REDACTED]

[5]

About the activation of KCNQ2/3 by SMIT1. Is the presence of SMIT1 molecule itself is important, or does the uptake of myo-IP also contribute? Some experimental data (e.g. comparison of the effect of SMIT1 on KCNQ2/3 channel, between (-) myoIP and (+) myoIP in the bath solution) are awaited.

>We have previously published this (Neverisky and Abbott, FASEB Journal 2017; Manville et al., Biophysical Journal 2017) - apologies for not making this clear. We showed that SMIT1 negative-shifts the voltage dependence of KCNQ2/3 activation, without the need for *myo*-inositol. On top of this, addition of myo-inositol has an additive effect because it increases local PIP2 that further augments KCNQ2/3 activity. We have added this explanation to the Introduction (page 4, final paragraph).

[6]

About the inhibition of SMIT1 by KCNQ2/3. Is the presence of KCNQ2/3 molecule itself is important, or does the ionic current flow and/or change of E_m also contribute? What will happen if non-conducting pore mutant is used?

>We previously published that the KCNQ2 non-conducting G279S pore mutant is equally effective as wild-type KCNQ2 at inhibiting SMIT1 (Neverisky and Abbott, FASEB Journal 2017).

[7]

The discussion about the negative feedback regulation (page 13) is very interesting, but I am afraid it is over discussion at this stage with very limited evidences.

>We have toned down this discussion in response to the reviewer's comment.

Reviewer #3 (Remarks to the Author):

Previous experiments carried out in the laboratory of the authors of this article had revealed the existence of a physical interaction between the SMIT1 myo-inositol transporter and the KCNQ2/3 potassium channel. In an observation that in principle did not seem related, the authors had determined that GABA was able to interact specifically with these potassium channels. Now the authors have observed that in DRG neurons, GABA modulates SMIT1 activity. The oocyte expression of the channel and the transporter seems to recapitulate the effect of the GABA observed in neurons. Using this experimental system, they have determined that the effect of GABA (and related metabolites) on SMT1 depends on the presence of the channel, which by binding GABA to its S5 domain is able to transmit the information to the attached transporter. In addition, this communication is bidirectional, since SMT1 alters the effect of GABA and related metabolites on the gating of potassium channels. Through directed mutagenesis they identify an arginine residue located in the channel voltage sensor as a

mediator of this interaction.

This part of the article represents a solid piece of work in which the existence of a dialogue between both proteins is proven. In a second part, perhaps more debatable, the authors try to raise the general issue of the existence of a crosstalk between potassium channels and transporters by including two additional examples. More than the hypothesis itself, which is attractive, because experimental support for the mechanisms of this crosstalk for these other proteins is weaker.

[REDACTED]

This reviewer has nothing against the experiments shown, but the characterization of the crosstalk is more superficial, and also the meaning of these interactions in a more physiological system is still uncertain.

[REDACTED]

REVIEWERS' COMMENTS:

Reviewer #2 (Remarks to the Author):

I evaluate the authors revised the manuscript intensively and satisfactorily.

(1) They changed the position of Fig 4(previous) to Fig 5(new). By this change, the flow and the readability became better.

(2) They toned down the over discussion about the negative feedback regulation.

(3) They completely omitted the previous Figs 6, 7 and related descriptions, which were weak part of the previous version. By this omission, the paper is now better focused. I wish to see in future the authors' next paper in which this topic is thoroughly studied.

I judge this paper contains important and interesting data. Due to the presence of many cases and parameters, the readability is still not perfect. However, I judge the way of presentation of the revised version is the practically possible acceptable choice.

I have no more specific comments.